# A New FPGA-Based Task Scheduler for Real-Time Systems

Lukáš Kohútka [1,*] and Ján Mach [2]

1  Institute of Informatics, Information Systems and Software Engineering, Slovak University of Technology in Bratislava, 812 43 Bratislava, Slovakia
2  Institute of Computer Engineering and Applied Informatics, Slovak University of Technology in Bratislava, 812 43 Bratislava, Slovakia
*  Correspondence: lukas.kohutka@stuba.sk

**Abstract:** This research demonstrates a novel design of an FPGA-implemented task scheduler for real-time systems that supports both aperiodic and periodic tasks. The periodic tasks are automatically restarted once their period has expired without any need for software intervention. The proposed scheduler utilizes the Earliest-Deadline First (EDF) algorithm and is optimized for multi-core CPUs, capable of executing up to four threads simultaneously. The scheduler also provides support for task suspension, resumption, and enabling inter-task synchronization. The design is based on priority queues, which play a crucial role in decision making and time management. Thanks to the hardware acceleration of the scheduler and the hardware implementation of priority queues, it operates in only two clock cycles, regardless of the number of tasks in the system. The results of the FPGA synthesis, performed on an Intel FPGA device (Cyclone V family), are presented in the paper. The proposed solution was validated through a simplified version of the Universal Verification Methodology (UVM) with millions of test instructions and random deadline and period values.

**Keywords:** real-time; task scheduling; EDF; FPGA; hardware acceleration; periodic tasks; CPU; SoC

## 1. Introduction

Real-time systems are a type of embedded system that handles tasks that require real-time processing. The success of these tasks is dependent on both the accuracy of the results and the time at which they are finished. Missing deadlines for real-time tasks can be considered a failure, just like calculating the wrong results. Therefore, real-time system reliability is achieved when tasks are finished within the specified time frame [1,2].

Task scheduling algorithms often use priority queues that are performed within software. Such software-based solutions work well for relatively simple and tiny real-time systems containing a limited number of tasks. However, as the number and complexity of tasks increase, the performance and constant response time become more critical, especially for safety-critical systems. Meeting the deadlines of tasks is considered a reliability requirement, as missing a deadline is seen as a failure of the system. Even using a high-performance processor does not guarantee that all tasks will meet their deadlines, which is why a dedicated task scheduler is necessary for real-time and safety-critical systems [3–7].

Real-time task scheduling is a critical aspect of many computing systems, particularly those that are safety-critical, time-critical, or both. Unfortunately, software-based solutions for task scheduling have several limitations, particularly with regards to the time consumed for scheduling, the determinism, dependability, and predictability of the system, and the algorithms used for scheduling. In an ideal world, a task scheduler for real-time systems would be able to generate an optimal schedule, using zero CPU time to perform the scheduling itself. This would allow all of the CPU time to be used for executing the tasks themselves rather than for scheduling. Of course, this ideal scenario is unlikely to be achievable in practice, and real-world schedulers will always spend CPU time to some extent. However, it is possible to minimize the time spent on scheduling and make

it as constant as possible, which would improve the determinism, dependability, and predictability of a real-time system. One drawback of task schedulers implemented in software is that they are restricted in the algorithms they can use, as they need to consume a very low and constant quantity of processor time. This often results in the use of priority-based scheduling algorithms, rather than algorithms based on task deadlines, leading to a lack of robustness and efficiency in the scheduling process. One possible solution to these limitations is the use of hardware acceleration for task scheduling. By implementing the task scheduler in hardware, it is possible to use deadline-based scheduling algorithms that minimize the time spent on scheduling and make it constant as well. This would allow for a more robust and efficient scheduling process, which would be a significant improvement over software-based solutions [7–16].

Hardware-accelerated task schedulers have been traditionally used for simple systems that consist solely of aperiodic hard real-time (RT) tasks. However, these solutions have proven to be insufficient for more complex and robust real-time systems that have a higher number of tasks and a greater variety of task types. To address this issue, a more advanced and sophisticated task scheduler is required. One that has the ability to support a diverse range of processes/threads. Apart from aperiodic RT tasks, hard RT systems also perform periodic tasks. These tasks can be managed by task schedulers in the same way as aperiodic tasks. However, including dedicated support for periodic tasks directly within the HW-based scheduler can significantly improve the overall performance of the system. This is because the addition of this support in HW eliminates the need for any further software extensions for periodic task management because periodic tasks are autonomously rescheduled without using the CPU whenever a period is completed. As a result, the system can operate more efficiently and effectively, providing the desired level of real-time performance and reliability. Overall, the implementation of a more robust and complex task scheduler is necessary to meet the demands of modern real-time systems that have a greater number of tasks and a wider range of task types. This will ensure that the system can perform at its optimal level and deliver the desired level of performance, reliability, and functionality [17–24].

The aim of the research presented in this article is to design a new version of a coprocessor unit that is capable of scheduling tasks based on the Earliest-Deadline First (EDF) algorithm [7]. The EDF algorithm is widely regarded as a dynamic version of deadline-based scheduling algorithms [25], as it eliminates the need for assigning individual task priorities. One of the main challenges faced by modern CPUs is the parallel execution of multiple tasks, which is a result of the widespread adoption of the many-core paradigm in processor design. This poses significant complications for hardware-accelerated task scheduling, particularly for RT systems that contain periodic hard RT tasks and require inter-task synchronization. In order to address these challenges, the research focuses on ensuring that the coprocessor unit is highly efficient and scalable in terms of performance while also ensuring that the overall system remains reliable and deterministic. This is a critical aspect that must be considered when developing hardware-accelerated task scheduling systems for real-time systems. In principle, the research presented in this article seeks to provide a solution to the obstacles and challenges faced by modern CPUs in the implementation of hardware-accelerated task scheduling. By focusing on the implementation of the EDF algorithm and addressing the critical aspects of efficiency, scalability, reliability, and determinism, the proposed coprocessor unit is designed to deliver improved performance and functionality for real-time systems containing periodic hard RT tasks.

This paper is structured the following way: The paper's Section 2 covers task schedulers for real-time systems. In Section 3, the paper introduces a new solution for task scheduling. The proposed solution is verified in Section 4. The paper presents synthesis results for the proposed solution in Section 5 and discusses the outcomes. Section 6 contains an evaluation of the performance achieved by the proposed solution, including a comparison with software-based scheduling. Finally, the paper concludes with a summary in Section 7.

## 2. Related Work

There are a lot of scheduling algorithms, each with pros and cons [26,27]. A deep comparison of global, partitioned, and clustered EDF scheduling algorithms in software has been presented in [28]. The authors performed experiments on a 24-core Intel Xeon L7455 system, where each core was running at 2.13 GHz. For these algorithms, they analyzed the overheads of the algorithms for the scheduling of tasks, their release, context switching, and several other types of overheads. The outcome was that as the number of tasks increases, the scheduling overhead is increasing, mainly for the global EDF, where the worst-case scheduling overhead for 250 tasks was 200 μs. The partitioned and clustered EDF algorithms reached up to around 30 μs overhead.

The scheduling overhead analysis of several algorithms has also been done in [29]. The authors run tests on a single-core ATmega2560 clocked at 16 MHz. The overhead of two tested non-preemptive EDF schedulers increased with the number of tasks. The maximum scheduling overhead by the basic EDF for twelve tasks was 136 μs, while the Critical-Window EDF had a maximum overhead of 404 μs.

Authors in [30] used the profiler of the Virtual Machine for a comparison of two algorithms to get information about their scheduling overhead on multiple cores. The first algorithm (LRE-LT) tried very hard to ensure that all deadlines were met. On the other hand, the second algorithm (USG) tried to minimize task preemption and migrations between cores, so it was expected that the second one missed a few deadlines. It was shown that the scheduler has been invoked a lot more often in the LRE-L and that the decision-making process took longer. The result was that the time spent on scheduling was approximately 10,000 times longer in the LRE-LT than in the USG.

Task scheduling plays a crucial role in operating systems, as it determines which task (i.e., thread or process) should be running in the processor and in what order. The algorithms used for scheduling greatly impact these decisions. Classic operating systems typically schedule tasks based on their priorities of tasks, while RT systems must schedule tasks based on their deadlines. This is because meeting the deadlines of all hard real-time tasks is of the utmost importance in real-time systems. The Earliest-Deadline First (EDF) algorithm is one of the most widely used and well-known algorithms for scheduling hard real-time tasks. It operates by sorting all tasks based on their deadlines, with the task having the earliest deadline being selected for execution first. Since tasks need to be sorted according to their deadlines, priority queues are the ideal data structure for implementing the EDF algorithm. Also, task scheduling is a core and critical component of operating systems that must be carefully designed and implemented [7,31,32].

The ideal real-time task scheduler is one that schedules tasks optimally, ensuring that all tasks are completed before their deadlines while minimizing the overhead on the CPU. The more CPU time that is consumed by the scheduling algorithm, the less effective the CPU becomes at executing the scheduled tasks. It is inevitable that some CPU time will be consumed, as the scheduler must use at minimum one clock cycle for transferring data to and from the scheduling unit. However, to achieve optimal performance, the goal is to spend the minimum amount of processor time. Maintaining a predictable and deterministic embedded system is critical, and this requires that a constant amount of CPU time be spent on scheduling, regardless of the actual amount of tasks currently present in the scheduler or even the maximum amount of tasks that can be handled by the system (i.e., the capacity of the task queue). This ensures that the system operates in a consistent and predictable manner, making it easier to debug and optimize. These qualities are essential for ensuring that real-time systems operate reliably and efficiently.

Our previous research [33] resulted in the development of a novel real-time task scheduler based on the Earliest-Deadline First (EDF) algorithm. This scheduler was implemented as a coprocessor unit, and a comparison was made between HW and SW realizations with regards to efficiency and performance. The coprocessor is designed to consume two cycles of the processor's clock domain, no matter how many tasks the system contains. Subsequently, an improved variant of the coprocessor was developed to be suitable for

dual-core CPUs. To solve the issue of conflicting situations where more than one CPU core attempts to access the coprocessor at the same time, two approaches were proposed and compared [34]. Finally, support for scheduling non-real-time tasks was added to the scheduler, utilizing priorities instead of deadlines [35].

In addition to our coprocessor, there are other existing solutions for HW-accelerated task scheduling on RT systems. Some of them are also utilizing the EDF algorithm [21,22,36,37]. One solution, presented in [37], uses the EDF algorithm as well, but with a limited maximum number of tasks of 64. On the other hand, another solution relies on task priorities instead of deadlines, which is not suitable for hard RT systems [38]. There are also other approaches that adopt a priority-based or static scheduling method [39–42]. One solution, presented in [43], supports EDF, LST (Least-Slack-Time) and priority-based scheduling. Schedulers based on genetic algorithms and fuzzy logic are presented in [44,45].

Our previous coprocessor solution is efficient for simple RT systems with hard RT tasks in conjunction with single-core CPUs. On the other hand, as RT systems become more complex and require higher performance, multi-core CPUs are often used, requiring a more complex task scheduler to support multiple cores. A thorough analysis of the suitability of the EDF algorithm in uniform multiprocessor systems reported that this algorithm is applicable in such cases too [37]. Some of the existing solutions also incorporate a method to monitor the remaining execution time of real-time tasks and predict potential deadline misses [38,41,42]. Based on the analysis of existing schedulers, we have decided to design a new RT task scheduler implemented as a coprocessor unit suitable for quad-core RT embedded systems and that would support periodic tasks and inter-task synchronization too.

The performance of real-time task schedulers based on deadlines relies heavily on the ability to sort tasks using their deadlines. To achieve this, a priority queue data structure is used as the central component in hardware-implemented task schedulers. There have been numerous designs for data sorting in priority queues for real-time systems, including the FIFO approach [36,38,40,46], Shift Registers [37,47–49], and Systolic Array [41,42,49,50]. Each of these architectures has been developed to provide efficient sorting capabilities, making them popular choices for use in real-time task scheduling.

The FIFO approach is extremely inefficient in terms of chip area and suitable for a small range of possible values (deadlines in this case), up to four or five bits only [36,38,40,46].

The architecture called Shift Registers is made up of homogeneous cells that each consist of a comparator, control logic, and registers to store one item. The cells are connected in a line, and each cell can exchange items with its two neighbors. The cells receive instructions simultaneously from the queue input, which results in an increase in the critical path length with an increase in the number of cells. The critical path length is a result of the bus width used for simultaneous instruction delivery and the exchange of control signals between all cells. The critical path issue of Shift Registers can be resolved by using a register at the inputs of cells, dedicating one clock cycle for the shared bus. The throughput of the Shift Registers architecture is one instruction per clock cycle. An illustration of a four-cell Shift Registers architecture can be seen in Figure 1 [37,47–49].

The architecture called Systolic Array is quite like Shift Registers, but it overcomes the critical path length issue by utilizing pipelining. The Systolic Array features homogenous cells, which are connected in a linear manner, with each cell having one neighboring cell to the left and right, excluding the first and last cell of the structure. The first cell in the queue serves as the only source of output for the entire queue and is receiving instructions through the queue's input. All instructions are gradually passed from one cell to the next cell, at a rate of one cell per clock cycle, in a manner similar to how instructions are processed through pipeline stages in pipelined processors. The throughput, however, of this architecture is smaller than Shift Registers because each delete instruction takes two clock cycles instead of just one. An example of the Systolic Array architecture consisting of four cells is displayed in Figure 2. The first cell on the right side contains the first item in

the priority queue and serves as the interface between the surrounding circuits. Clock and reset are the only parallel signals [41,42,49,50].

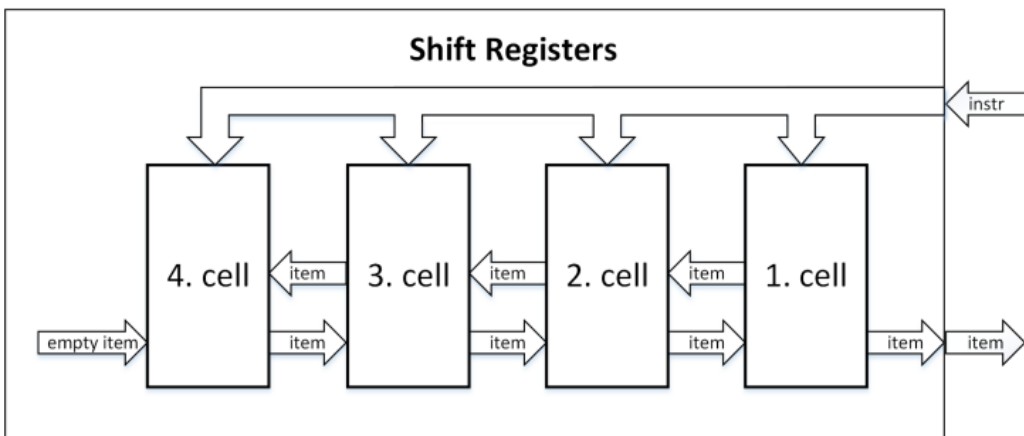

**Figure 1.** Shift Registers architecture example [48].

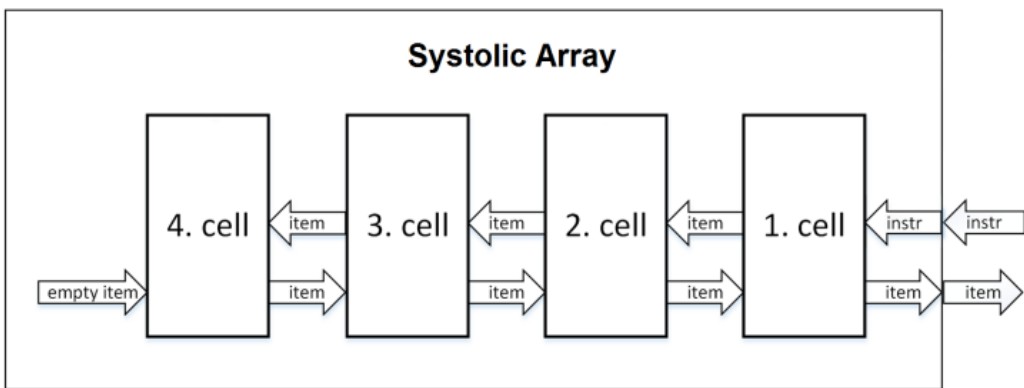

**Figure 2.** Systolic Array architecture example [41,42,49,50].

The Systolic Array priority queue brings a solution to the critical path issue present in Shift Registers. The instructions are propagated through the cells, with each cell representing one pipeline stage with pipeline registers. This means N clock cycles are required for an instruction to propagate via the entire queue, where N is the queue capacity. Since every cell is performing a different instruction at a time, the throughput of this structure is one instruction per clock cycle. However, after deleting an instruction, a pause (NOP) is needed for one cycle, resulting in a throughput of one instruction per two clock cycles. The priority queue is providing an updated output in just two cycles. Thus, response time is two clock cycles. This response time remains constant, regardless of the number of cells in the queue. In addition to the lower throughput of this architecture in comparison to Shift Registers, the second disadvantage is the almost doubled amount of flip flops needed for implementation of the Systolic Array [41,42,49,50].

## 3. Proposed Solution

The proposed solution is a task scheduler based on FPGA technology, which is designed as a coprocessor that receives instructions from the processor and sends back decisions about which instructions are supposed to be executed at the moment. This implies that the scheduler will be encapsulated or integrated into an existing CPU, much like any other coprocessor, for example a multiplier or divider. Figure 3 illustrates the architecture at the top level of abstraction of the top-level module of the designed coprocessor, which is comprised of seven submodules: Ready Queue, Waiting Queue, Idle Queue, Tasks Memory, Running Tasks, Control Unit, and Semaphore.

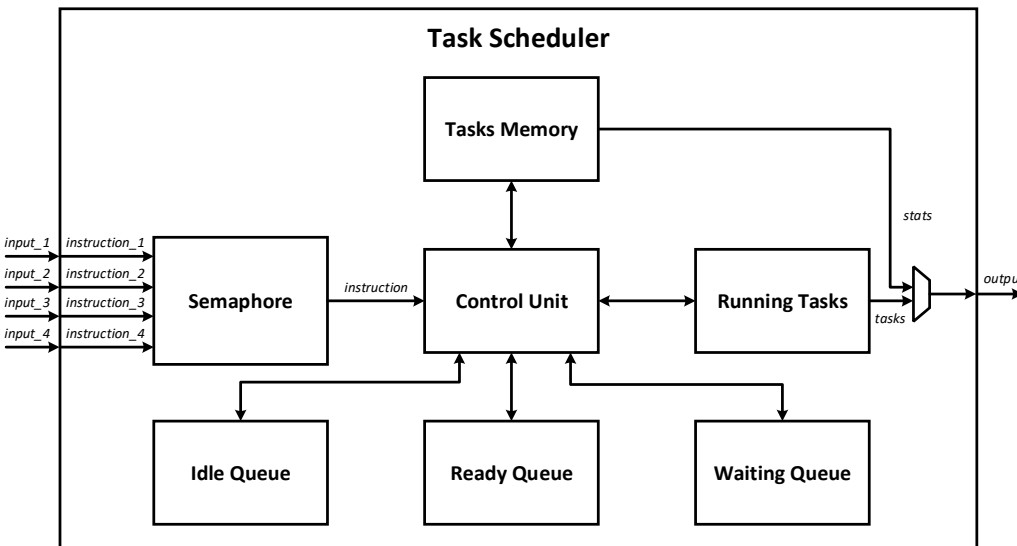

**Figure 3.** Block diagram of proposed task scheduler.

The scheduler top-level module contains four input ports, known as instr_1, instr_2, instr_3 and instr_4, which are used by the CPU to provide the coprocessor instructions for the scheduler. It is assumed that up to four tasks/processes/threads can run simultaneously (for example, on a quad-core CPU). The output port of the scheduler is providing valuable data for the processor. Using this output port, the processor also gets the information about which (up to four) processes are currently dedicated to run right now, and the processor is also able to obtain memory data from the Tasks Memory submodule this way as well.

The proposed task scheduler is providing the following list of new instructions:

- MEMORY_WRITE—can be used to create a new task in Tasks Memory or to modify already created tasks. This instruction performs a standard write operation into the memory.
- MEMORY_READ—can be used to read any information about tasks stored in Tasks Memory. This instruction performs a standard read operation from the memory.
- SCHEDULE_TASK—is used to schedule an existing task to be executed by the CPU. This causes the task to be moved to Running Tasks or Ready Queue, which is a decision based on the scheduling algorithm (i.e., deadline values). Task states that are stored inside Tasks Memory will be modified if needed, too.
- KILL_TASKL—is used to deschedule (i.e., to kill) an already scheduled task. As a result, the task is removed from the queues, such as Running Tasks, and the task state is set to IDLE_TASK in Tasks Memory.
- BLOCK_TASK—is used to temporarily block a scheduled task, forbidding its execution for a limited time. As a result, the state of the selected task is changed to WAITING, and the task is moved into the Waiting Queue. The task is blocked for a specified time only; therefore, a waiting time is set too. When the waiting time elapses, the blocked task will be automatically unblocked.
- UNBLOCK_TASK—is used to unblock a blocked task. As a result, an existing blocked task is released (i.e., unblocked), changing its state from WAITING to a different state, and this task is also removed from the Waiting Queue, returning the task back to the Ready Queue or Running Tasks. Since blocked tasks are automatically unblocked after a specific waiting time elapses, this instruction is just meant for unblocking the task earlier, eliminating the need to wait until the waiting time has elapsed.
- GET_RUNNING_TASKS—is used to obtain the list of running tasks or the task that is selected for execution in a particular processor core using the scheduler output port. This information is provided by the Running Tasks module.

### 3.1. Ready Queue

The Ready Queue is a key component in the task scheduler, as it holds all tasks that are ready for execution but are waiting their turn. This component is designed as a priority queue, sorting tasks that are ready by their deadline values so that the next task to be executed can be quickly identified. The sorting of tasks is performed by utilizing the Shift Registers architecture, which was explained in Section 2. This architecture is composed of sorting cells, each of which consists of a comparator for comparing deadlines, control logic for deciding when and what to store in this cell, and a register as an actual storage element to remember the ID of the task and its deadline. These cells are able to move tasks to neighboring cells, and they receive instructions simultaneously via a common bus. Ready Queue ensures that the tasks are executed in a timely manner, based on their deadlines, and provides the necessary information to the CPU for task selection and execution.

### 3.2. Waiting Queue

The Waiting Queue is an integral part of the task scheduler and is designed to hold all the temporarily suspended or blocked tasks. This component is also implemented as a priority queue, just like the Ready Queue, to sort the waiting tasks based on their remaining waiting times. The waiting tasks are referred to as such because they are temporarily put on hold and are waiting to be unblocked. This can occur in two ways: either through the execution of the UNBLOCK_TASK instruction or if the remaining time for waiting has elapsed. Once a task is unblocked, it is extracted from the Waiting Queue. This prioritization of waiting tasks ensures that the task scheduler can effectively manage the execution of multiple tasks and maintain a high level of performance.

The Waiting Queue is used only for the inter-task synchronization instructions BLOCK_TASK and UNBLOCK_TASK. While this queue is allowing users to block and unblock tasks, the inter-task synchronization logic itself is not implemented and is only supported by providing these two instructions. It is up to the software extension to decide whether and when a particular task is supposed to be blocked, for how long it is blocked, and eventually, whether a blocked task is unblocked before the block time elapses.

### 3.3. Idle Queue

The Idle Queue is a module designed to hold idle periodic tasks (i.e., state = idle), either because they were completed naturally or terminated using the KILL_TASK instruction. This component is only relevant for periodic tasks; tasks that are not periodic are not stored in the Idle Queue. The module is structured as yet another priority queue, similar to Waiting Queue and Ready Queue, with each task being sorted based on their remaining period times. The output of Idle Queue represents the next periodic task that is going to finish its period. When current time reaches the task's period time, the task is extracted from the Idle Queue and rescheduled to start a new instance of this task for the new period. It is efficient handling of periodic tasks, and it is possible despite the fact that only one task can be extracted from the Idle Queue at a time. If more tasks end their period at the same time, i.e., they need rescheduling simultaneously, then those tasks that are rescheduled later automatically adjust their remaining deadline time by the delay incurred. Thus, such tasks may be rescheduled in any sequence under the condition that the rescheduled tasks get adjusted for their remaining deadlines appropriately.

Unlike the Waiting Queue, the Idle Queue does not need any instructions to be called from the CPU in order to manage the idle (completed) periodic tasks that are waiting for their next period. Whenever the period of the periodic task elapses, the task is automatically moved from the Idle Queue back to Running Tasks. The state of this task is automatically changed from idle to ready or running (depending on the EDF logic, i.e., deadlines of tasks). This automation is provided by the Control Unit module.

*3.4. Semaphore*

The Semaphore component is designed to handle conflicts that arise when multiple CPU cores simultaneously attempt to use the scheduler, e.g., to schedule a new task or to kill a task in exactly the same clock cycle. This situation is referred to as a conflict. To resolve conflicts, the Semaphore is a module that is responsible for arbitrating instructions by selecting one of the instructions as the arbitration winner and the rest of the instructions becoming losers. The winner's instruction is passed on to the Control Unit. The loser instructions cannot be executed immediately; therefore, other CPU cores are stalled. Semaphore module uses a widely known algorithm called Round-Robin, which guarantees that the arbitration is fair, and the load is evenly balanced. This is an important aspect of the FPGA-based task scheduler design, as it ensures that all CPU cores have equal access to the scheduler coprocessor and that no core is favored over the others.

The total number of possible conflicts that can arise in the system is eleven, and each conflict is identified by a different combination of CPU cores attempting to use the scheduler at the same time. When two processor cores try to access the coprocessor simultaneously, there are six combinations possible: 1_2 (processor core 1 and core 2 conflict), 1_3 (core 1 and core 3 conflict), 1_4 (core 1 and core 4 conflict), 2_3 (core 2 and core 3 conflict), 2_4 (core 2 and core 4 conflict), and 3_4 (core 3 and core 4 conflict). In the case where three CPU cores attempt to access the scheduler at the same time, there are four possible combinations: 1_2_3 (core 1, core 2, and core 3 conflict), 1_2_4 (core 1, core 2, and core 4 conflict), 1_3_4 (core 1, core 3, and core 4 conflict), and 2_3_4 (core 2, core 3, and core 4 conflict). The final combination occurs if all four processor cores try to access the coprocessor simultaneously, and this is named 1_2_3_4.

The Semaphore module has two essential demands, a primary and a secondary one. The most critical demand is to have a limited maximum number of delays caused by conflicts and to keep this number low. This demand is vital since the scheduler is designed for RT embedded systems. The second demand is to have fairness among all CPU cores, ensuring that each core has a similar chance of winning the conflict and accessing the scheduler immediately.

The design of the Semaphore module is based on a 2-bit counter to represent four distinct states. These states, referred to as 1234, 2143, 3412, and 4321, are used to determine the priority order in case of any of these eleven possible combinations of access conflicts. For instance, if the current state is 1234, then processor core 1 is prioritized over core 2, processor core 2 is prioritized over core 3, and processor core 3 is prioritized over core 4. To resolve the conflicts, the current state is shifted to the new state through an increment of a 2-bit counter. The decision was taken to limit the amount of possible priority order permutations to just four instead of the original twenty-four permutations in order to simplify the FSM that is deciding which processor core wins the arbitration. This results in a relatively simple design of the FSM consisting of four states instead of twenty-four states, reducing chip area and energy consumption. These four states have been carefully selected to ensure symmetry, fairness, and rotations after each conflict. Whenever there is a conflict, the order is updated by shifting FSM to the next state. The 2-bit counter is used to represent the states in the following manner:

- Counter set to "00" is representing state 1234, and is incremented to "01".
- Counter set to "01" is representing state 2143, and is incremented to "10".
- Counter set to "10" is representing state 3412, and is incremented to "11".
- Counter set to "11" is representing state 4321, and is incremented to "00".

The scenarios in which two or more CPU cores conflict, along with the winner selection, are presented in Table 1. This table outlines the different potential scenarios for conflicting CPU cores, with each line representing a possible conflict. The columns represent the four possible orders, where one order is used at the moment based on which state the FSM is currently in. It is noticeable that the primary as well as secondary demands for the Semaphore were satisfied, as each processor core stalled three times in a row, at most, and the distribution of wins among the CPU cores is even. The rotating nature of the

states/orders ensures that the worst-case scenario for one instruction to be completed is *2N* clock cycles (in the event that all processor cores attempt to access the coprocessor continuously), where *N* is the number of processor cores (i.e., *N* = 4). On the other hand, the best-case access time is just two cycles. Consequently, the amount of time it takes for one instruction to be executed on a quad-core CPU ranges from two to eight clock cycles, which depends on the frequency of access conflicts.

**Table 1.** Table of conflict resolutions in Semaphore module.

| Title 1 | 1234 | 2143 | 3412 | 4321 |
|---------|------|------|------|------|
| 1_2 | 1 | 2 | 1 | 2 |
| 1_3 | 1 | 1 | 3 | 3 |
| 1_4 | 1 | 1 | 4 | 4 |
| 2_3 | 2 | 2 | 3 | 3 |
| 2_4 | 2 | 2 | 4 | 4 |
| 3_4 | 3 | 4 | 3 | 4 |
| 1_2_3 | 1 | 2 | 3 | 3 |
| 1_2_4 | 1 | 2 | 4 | 4 |
| 1_3_4 | 1 | 1 | 3 | 4 |
| 2_3_4 | 2 | 2 | 3 | 4 |
| 1_2_3_4 | 1 | 2 | 3 | 4 |

A block diagram of the Semaphore module is shown in Figure 4, based on the previously described information. The module includes 3 multi-bit multiplexers, 2 D-FFs, 5 AND gates, 4 NAND gates, 2 inverters, a Winner Selector submodule, and a Conflict Detector. The Winner Selector and Conflict Detector submodules only require the instruction's bit, which indicates if the instruction is valid or not (if the processor is attempting to access the coprocessor). The Winner Selector submodule selects the winner of arbitration, i.e., deciding which of the valid instructions should be selected. The selection signals are called SEL00, SEL01, and SEL1. These signals drive the control inputs of MUXes that pass the winning instruction to an output port named "instr". This port is used by Control Unit module.

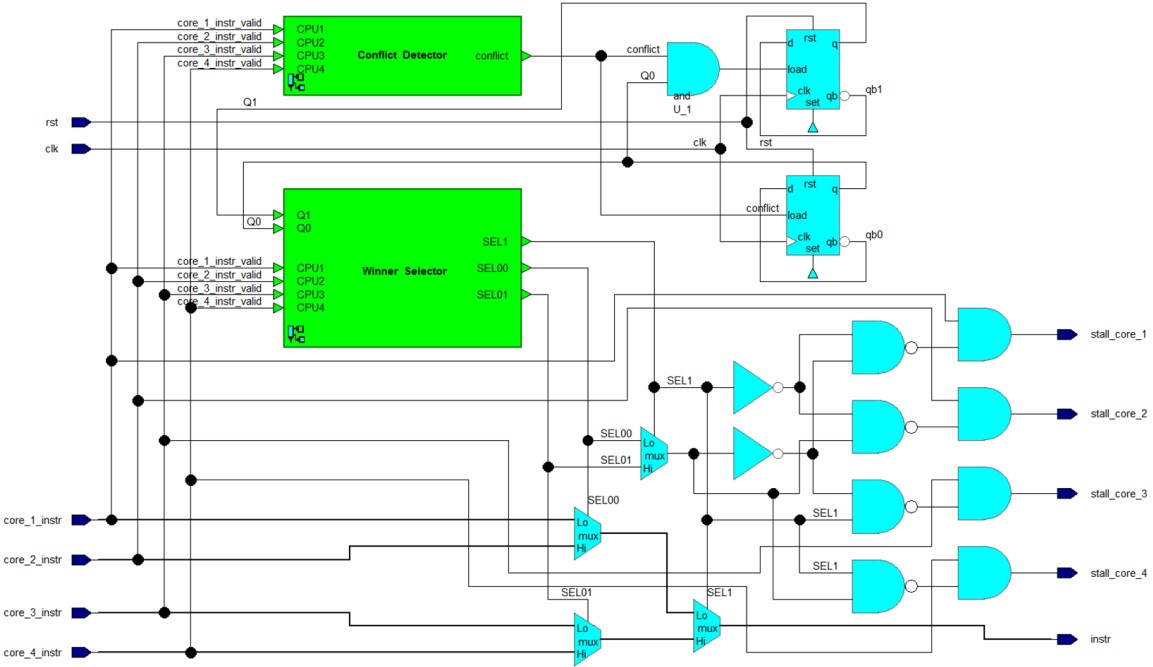

**Figure 4.** Block diagram of Semaphore module.

The Semaphore module in Figure 4 also outputs four single-bit outputs to the individual processor cores, named stall_core_#. These signals serve to notify the respective processor core when the attempt to access the coprocessor is declined due to losing an arbitration caused by the situation when another core is trying to use the coprocessor as well (i.e., during a conflict). The processor core receiving the stall response must wait until the stall is active. In the meantime, it can perform other operations. The signals core_#_instr_valid and stall_core_# act as a means of handshaking-based communication between the processor and coprocessor.

The Conflict Detector submodule is illustrated in Figure 5 as a logic circuit. It consists of one 6-input NAND gate and six basic 2-input NAND gates. The purpose of this submodule is to detect whether 2 or more processor cores are trying to access the coprocessor simultaneously. If two or more inputs are set to 1, the conflict output will be 1 to indicate a conflict. However, if there is only one valid instruction from a single CPU core at most, the conflict output will be 0.

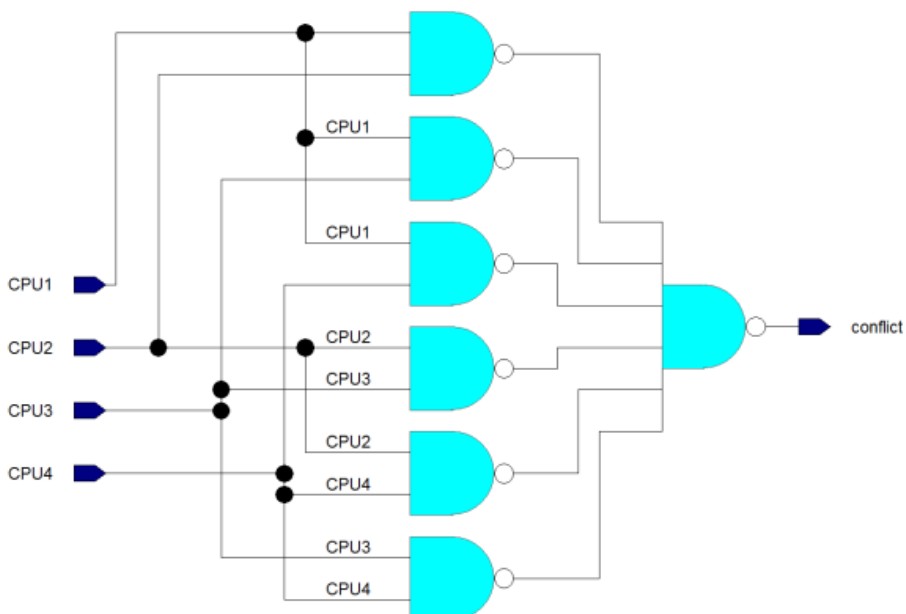

**Figure 5.** Logic circuit of Conflict Detector.

The Winner Selector submodule, as depicted in Figure 6, implements its decision logic through the use of one 3-input NOR and three 2-input NORs for the SEL1 output, and two 2-input NANDs for each of the SEL00 and SEL01 outputs. The circuit includes two inverter gates for creating inverted input signals as well. The decision logic for this module is the same as that outlined in Table 1. The Q1 and Q0 inputs represent the actual state of the FSM. The inputs CPU1, CPU2, CPU3, and CPU4 represent information about which processor cores are attempting to access the coprocessor at the moment. The output signals SEL00, SEL01, and SEL1 are needed for controlling the multiplexing presented in Figure 4.

*3.5. Running Tasks*

The Running Tasks module holds the tasks that are currently being executed. With a capacity of four tasks, it can execute up to four tasks at once. To minimize unnecessary task switches, this component can assign a task that was previously running on one CPU core to another CPU core due to task preemption. By doing so, the number of task switches is reduced to the minimum possible amount, and the scheduling overhead is minimized this way.

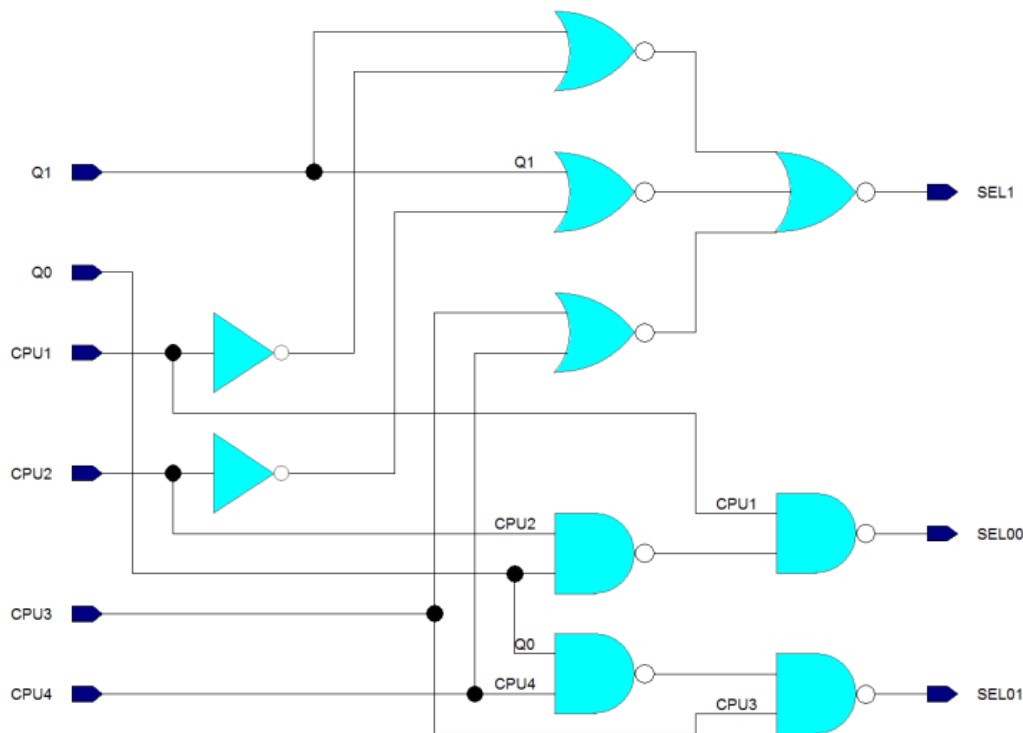

**Figure 6.** Winner Selector logic circuit.

The control logic of Running Tasks performs decisions about whether to keep the current tasks or make changes. If a task is terminated, the task that has the lowest deadline value within the Ready Tasks submodule is moved into the Running Tasks submodule. When the coprocessor is scheduling new tasks, a preemption may occur, depending on the task deadline and the deadlines of currently running tasks. Whenever the task is scheduled with an earlier deadline than the deadline of any running task, the running task with the latest deadline is replaced, causing task preemption, and execution of the preempted task is suspended. If preemption occurs, the preempted task is stored in the Ready Tasks module; otherwise, the new task is added to it.

The Running Tasks component has five comparators and requires two clock cycles for decision logic due to the length of the critical path in the combination logic. The results of the comparison performed in the first cycle are stored in registers holding temporary results along with 2 bits for task identification. The first cycle performs two parallel comparisons, comparing running_task_core_1 with running_task_core_2 and running_task_core_3 with running_task_core_4. During the second cycle, the deadlines of temporary results obtained from the registers are compared, which may result in a preemption, replacing one running task with a new one. The previously running task that is being replaced is sent to the Ready Queue if preemption occurs. Regardless of preemption, one of the tasks is sent to the Ready Tasks module—either the new task (i.e., no preemption occurs) or one of the previously running tasks (i.e., preemption occurs).

### 3.6. Control Unit

The Control Unit component manages all task queue modules (Idle Queue, Waiting Queue, Ready Queue, and Running Tasks) and reads from and writes to the Tasks Memory module. When a valid instruction is received from the Semaphore component, the Control Unit module decodes it and performs it by providing data and control signals to surrounding modules. The Control Unit directly controls all task queue components, except for the Ready Queue, which it can only access indirectly using the Running Tasks module. When no valid instruction is received from the CPU or Semaphore, the Control Unit can still transfer tasks autonomously between the Running Tasks module and Idle Queue module

and between the Running Tasks module and Waiting Queue module. This ensures that managing waiting or blocked tasks, including periodic idle tasks, is done automatically, reducing the CPU time needed for task scheduling, which is crucial for RT tasks.

Control Unit is also responsible for managing states of tasks. There are four possible states the task can have in this scheduler: IDLE, RUNNING, READY and WAITING. The IDLE state is used for tasks that are not yet scheduled or are completed already. This is especially important for periodic tasks to determine that the task is finished, and that the scheduler is waiting for the next period of the task to automatically schedule a new instance of the periodic task. The RUNNING state is used for tasks that are running, i.e., being executed at the moment. The READY state is used for tasks that are scheduled and ready to be executed but have not yet been chosen for execution due to other tasks being prioritized over the ready task. WAITING state is used for those tasks that are blocked by the BLOCK_TASK instruction, so these tasks are waiting to be unblocked either by the UNBLOCK_TASK instruction or by elapsing the waiting time set by the BLOCK_TASK instruction. These states and possible changes between the states are depicted in Figure 7.

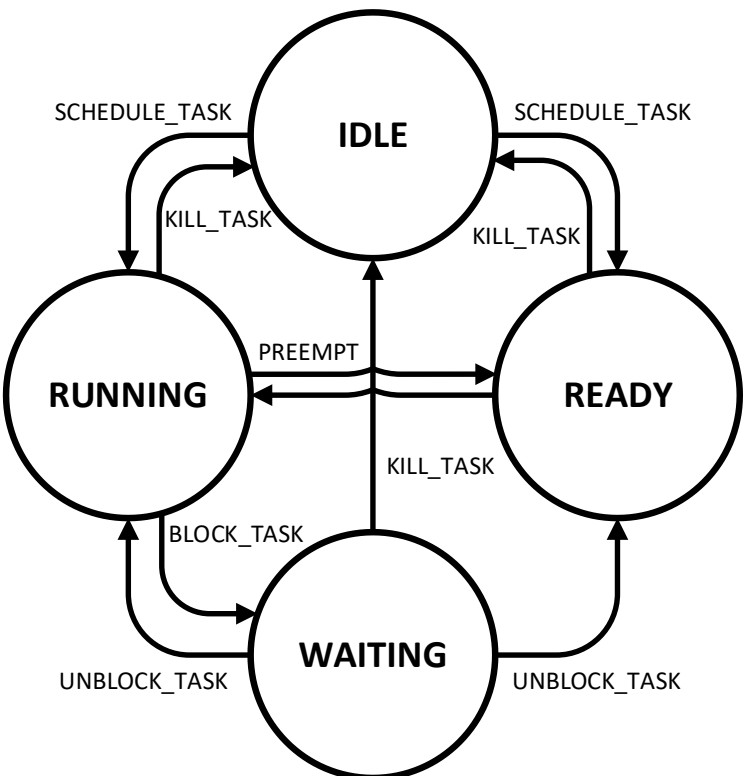

**Figure 7.** State diagram of task states.

### 3.7. Tasks Memory

The Tasks Memory component is a multi-port standard memory designed to support various features in other components. Though it could potentially be implemented using SRAM, it has been realized with registers due to the need for multiple read/write ports. While this implementation based on registers consumes more chip area, the added read/write ports are essential to the functions described in other components.

The Tasks Memory stores all information about tasks, including task type, task state, ID of parent task, and timing characteristics such as starting/remaining deadline, starting/remaining period (if the task is periodic), and starting/remaining execution time of the task. While the starting timing characteristics are provided by the CPU when a task is created, the remaining timing characteristics are automatically maintained by the scheduler itself. The memory's layout is outlined in Table 2, with the lowest three bits of address

being reserved to choose specific data within the particular task while the upper bits are utilized to choose a task.

**Table 2.** Memory map of Tasks Memory.

| Address Bits 2 Downto 0 | Field | Number of Bits |
|---|---|---|
| 000 | ID of parent task | 8 |
| 001 | Task state + task type | 4 + 5 |
| 010 | Remaining deadline time | 20 |
| 011 | Remaining period time | 20 |
| 100 | Remaining execution time | 20 |
| 101 | Starting deadline time | 20 |
| 110 | Starting period time | 20 |
| 111 | Starting execution time | 20 |

## 4. Design Verification

The task scheduling coprocessor that was introduced was described using the SystemVerilog language and then tested through simulations. The ModelSim tool was used to perform these simulations. Additionally, to the SystemVerilog language, a simplified variant of the Universal Verification Methodology (UVM) was also utilized during the verification phase. The interface of the scheduler was quite simple, making it possible to simplify the use of UVM as well. In this scenario, a single transaction within the UVM test equates to one instruction executed in two clock cycles, eliminating the need for agents to interface with the device under test (DUT).

The verification phase involved the use of one test procedure that generated constrained random input values, a scoreboard, and a predictor. This test generated millions of instructions with deterministic instruction opcodes and unique task ID values, but with random timing values. The predictor is responsible for predicting the expected output of DUT (Design Under Test) based on the input values. The predictor behaves similarly to the DUT but at a higher level of abstraction, similar to high-level software languages. The predictor's description was purely sequential and high-level, utilizing the SystemVerilog priority queue data structure and the sort() procedure to order the tasks within each queue. Figure 8 demonstrates the testbench that was applied for verification simulations.

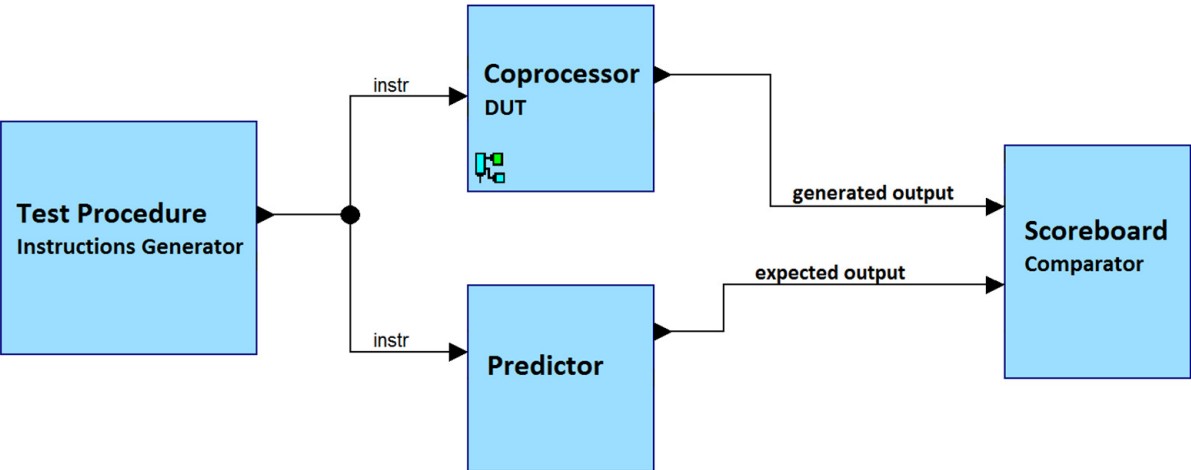

**Figure 8.** Testbench architecture.

To verify that the designed scheduler is working properly and as expected, more than 2,000,000 iterations of the test were performed, each containing at least 1000 randomly generated instructions. The full capacity of the scheduler was utilized during this test. Scheduler parameters were set to the following values during the design verification:

eight bits for task IDs, a capacity of Ready Queue set to sixty tasks, and twenty bits for random deadlines, execution times, and task periods.

The verification process was thorough, ensuring that the proposed task scheduler was functioning as expected. The use of the SystemVerilog language and UVM, along with the test procedure and predictor, provided a comprehensive and efficient method for verifying the coprocessor unit's behavior. The results of these simulations demonstrate the reliability and effectiveness of the task scheduler, making it a suitable solution for real-time task management.

## 5. Synthesis Results

The proposed task scheduler was implemented on an Intel Cyclone V FPGA, more specifically the 5CSEBA6U23I7 device. The synthesis process was performed using the Intel Quartus Prime 16.1 Lite Edition tool. To ensure that the scheduler would operate properly, a static timing analysis was performed to determine the maximum clock frequency for each version of the design.

The synthesis results presented in Table 3 indicate that the maximum clock frequency of all versions of the proposed scheduler is 105 MHz or higher. The critical path, i.e., the path that is limiting the maximum clock frequency, was found in priority queues. Therefore, increasing the size of these priority queues has an impact on the maximum clock frequency (fMax). However, the resource requirements, as measured by Adaptive Logic Module (ALM) consumption, are relatively low considering the large capacity of current FPGA devices, which often have hundreds of thousands, if not millions, of ALMs.

**Table 3.** FPGA synthesis results.

| Tasks Capacity | ALMs | Registers | fMax (MHz) |
|:---:|:---:|:---:|:---:|
| 8 | 334 | 325 | 177.99 |
| 16 | 591 | 541 | 156.98 |
| 24 | 832 | 756 | 137.85 |
| 32 | 1067 | 976 | 127.67 |
| 40 | 1324 | 1181 | 122.05 |
| 48 | 1576 | 1403 | 118.10 |
| 56 | 1817 | 1624 | 107.49 |
| 64 | 2044 | 1833 | 105.52 |

It is important to note that the Tasks Capacity, or the maximum number of tasks the scheduler can handle, has a direct and significant impact on both the resource costs and the maximum clock frequency. As the Tasks Capacity increases, the logic utilization increases and the timing performance (fMax) decreases. The implementation of each timing variable, such as deadline, period, waiting time, and execution time, uses twenty bits, while the task ID is comprised of eight bits.

## 6. Performance Evaluation

This section demonstrates the performance benefits of using the proposed task scheduler instead of the existing software-based task scheduler, the G-EDF (Global Earliest Deadline First) algorithm that was presented in [28] and used on a 24-core Intel Xeon CPU running at 2.13 GHz.

Two use cases of the proposed scheduler are considered: one case when a CPU that is running four tasks in parallel (i.e., four CPU cores) is used, and the second case when a CPU with one task (i.e., one CPU core) is used. In both cases, the proposed scheduler is running on the FPGA described in the previous section at 100 MHz. Using this clock frequency means that one clock cycle takes ten nanoseconds, which is equal to 0.01 us. Table 4 shows the worst-case CPU overhead of task scheduling, i.e., the time needed to schedule one task (to call one SCHEDULE_TASK instruction). This overhead is displayed in microseconds (us). The four CPU cores version takes seven clock cycles in the worst-case

scenario, which occurs when the CPU core has to wait for six clock cycles plus one clock cycle for calling the SCHEDULE_TASK instruction.

**Table 4.** Worst-case CPU overhead of task scheduling comparison in microseconds (us).

| Number of Tasks | G-EDF on Xeon [28] | Proposed Scheduler (with 4 CPU Cores) | Proposed Scheduler (with 1 CPU Core) |
|---|---|---|---|
| 25 | 20 | 0.07 | 0.01 |
| 36 | 28 | 0.07 | 0.01 |
| 50 | 42 | 0.07 | 0.01 |
| 64 | 51 | 0.07 | 0.01 |
| 100 | 140 | 0.07 | 0.01 |

The worst-case overhead of software-based EDF scheduling is around 20 us when 25 tasks are used. If this scheduling is hardware-accelerated using the proposed solution, then the overhead drops to less than 0.1 us, effectively reducing the CPU overhead more than 200-times, i.e., more than 99.5% overhead reduction is achieved. It is worth noting that the overhead of the proposed solution is constant with respect to the number of tasks—unlike in software-based scheduling, where the CPU overhead increases with the number of tasks. Thus, the relative reduction of CPU overhead is even 99.95% (i.e., 2000-times lower overhead) when 100 tasks are used. Therefore, the proposed HW-based scheduler has much better scalability for the growing number of tasks, allowing more complex real-time systems with a higher number of tasks to be implemented.

The proposed solution significantly outperformed the existing software-based scheduling despite the fact that the existing solution used a 2.13 GHz clock and the proposed solution used only a 100 MHz clock. If the proposed solution was implemented using cutting-edge ASIC technology together with a CPU, then the performance benefits would be even bigger, reducing the scheduling overhead down to around 3.5 ns (i.e., 0.0035 us) and further reducing the overhead by around 200-times if a 2 GHz clock was used. On the other hand, since the main limitation of the proposed solution is the amount of HW resources (i.e., chip area in ASIC or Look-Up Tables in FPGA), which heavily depends on the number of tasks to be supported, the FPGA technology brings a significant advantage in the form of configurability and reconfigurability (including the partial reconfiguration feature of FPGAs) of the scheduler. In FPGA, the scheduler can be configured to have optimal task capacity, whereas in ASIC, the scheduler cannot be reconfigured for optimal task capacity, which can lead to significant waste of chip area if the actual real-time system is using much less tasks than the scheduler capacity. The optimal setting of scheduler capacity is hard to determine as it depends on the real-time system requirements and needs.

The overall performance benefits of using a hardware-accelerated task scheduler also depend on what percentage of the CPU time is spent on the scheduling and what percentage is used for the actual execution of scheduled tasks. This depends on the actual application of the real-time system and the granularity of tasks—whether the application uses a few big tasks or many smaller ones. However, regardless of the actual application, by accelerating the task scheduling in hardware, it is possible to use almost all of the CPU time for the actual execution of scheduled tasks instead of scheduling those tasks. Thanks to Moore's Law, the costs of hardware-accelerated scheduling are gradually lower and lower, causing the overall benefits to outweigh the costs.

## 7. Conclusions

The proposed task scheduler is a novel solution that implements the Earliest-Deadline First (EDF) scheduling algorithm on an FPGA. This scheduler is well-suited for complex real-time systems that consist of a mixture of aperiodic hard RT tasks, periodic hard RT tasks, and non-real-time (best-effort) tasks. It leverages a priority queue-based approach to handle all types of tasks efficiently and with ease.

The scheduler uses priority queues not only to sort the ready tasks but also to handle idle periodic tasks and waiting/blocked tasks. As a result, managing these types of tasks is straightforward, allowing for an autonomous handling process with no need for software extension, with the only exception being the Tasks Memory initialization. Additionally, the priority queue-based approach makes the proposed scheduling solution highly efficient in managing periodic RT tasks and readily extensible to include task synchronization and inter-task communication capabilities.

This scheduler is optimized for use on quad-core CPUs, which can execute up to four tasks in parallel. All of the supported instructions take a maximum of three clock cycles to complete, no matter the system configuration or the actual or maximum amount of tasks in the system, provided there are not any conflicts between multiple processor cores attempting to access the scheduler simultaneously. However, in the event of such conflicts, an extra delay of two to six cycles may occur, leading to a maximum latency of nine clock cycles per instruction in the worst-case scenario.

In conclusion, the proposed task scheduler is a highly efficient solution for real-time systems that can handle a diverse range of tasks, including aperiodic hard RT tasks, periodic hard RT tasks, and non-real-time (best-effort) tasks. Its utilization of priority queues simplifies the handling of periodic idle tasks and blocked/waiting tasks, making it a suitable option for systems that require minimal software intervention. The scheduler's performance is further enhanced by its compatibility with quad-core CPUs and its ability to execute instructions in a few cycles, regardless of the number of tasks the system contains. These features make the proposed task scheduler an attractive option for complex real-time systems that require efficient task management and optimal performance.

**Author Contributions:** Conceptualization, L.K.; methodology, L.K.; software, L.K.; validation, L.K.; formal analysis, L.K.; investigation, L.K.; resources, L.K. and J.M.; data curation, L.K.; writing—original draft preparation, L.K.; writing—review and editing, L.K. and J.M.; visualization, L.K.; supervision, L.K.; project administration, L.K.; funding acquisition, L.K. All authors have read and agreed to the published version of the manuscript.

**Funding:** The work reported here was supported by the Operational Programme Integrated Infrastructure for the project Advancing University Capacity and Competence in Research, Development and Innovation (ACCORD) (ITMS code: 313021X329), co-funded by the European Regional Development Fund (ERDF). This publication was also supported in part by the Slovak national project KEGA 025STU-4/2022, APVV-19-0401 and APVV-20-0346.

**Data Availability Statement:** Not applicable.

**Conflicts of Interest:** The authors declare no conflict of interest.

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
