# Peer review of "A New FPGA-Based Task Scheduler for Real-Time Systems"

_electronics, doi:10.3390/electronics12081870_

Round 1
Reviewer 1 Report
Hello,
First of all I would like to congratulate the authors on their fine work.
The problem of RT task scheduling is properly approached in the paper. The use of a hw implementation is very welcomed and necessary in this case. I personally did not detect any gap in knowledge during the scientific process.
The cited references range from recent ones (less than 5 yr old) to older ones (more than 20 yr old) but taking in to account the fact that the researched problems were actual in the past 2 decades, this does not seem to be a problem.
If the specific HDL code were supplied I think that the manuscripts results would be reproductible using other FPGA products for testing, myself having access to AMD Xilinx products for testing.
The figures and tables are clear and on point, including sufficient data and a re clutter-free. Moreover, they are easy to understand an interpret.
The final conclusions are supported by the simulation data obtained using Modelsim, and the batch of iterations performed (2 000 000) seems appropriate.
The English language used in the presentation is correct, accurate and clearly presents the processes and components involved in the proposed hw design. There is small typos at line 123 where I would replace "cane" with can.
I strongly recommend publishing the paper due to the quality of the scientific effort and the obtained results.
Author Response
Thank you very much for your positive feedback.
Reviewer 2 Report
The authors describe how to do periodic and aperiodic task scheduling efficiently. This paper showed that considering the number of clock cycles, this methodology can be very useful for real-time complex task scheduling problems and can provide optimal solution.
Pros:
1) The paper is well motivated. It attacks a very relevant problem in an operating system which tsaks should be running and the schedule should be in which order. These authors have used EDF algorithm to schedule task. This paper used SystemVerilog language to implement this algorithm and used ModelSim tool to perform these simulations and verified their results.
Cons:
1) There is not enough comparison how their proposed feature gives better result compared to their previous work and other algorithms.
Comments:
- This paper is well written. The block diagram of task scheduler gives the whole design in a high level where inputs, outputs and seven submodules are shown. The authors also explain very well the reason behind choosing seven features and what is their impact. Their test generated random timing values and millions of instructions with deterministic instruction opcode and unique task ID values. The hardware implementation result is also given with the frequency. One spelling mistakes in page 3(cane will be can)
Overall, I like the paper.
Author Response
Thank you very much for your positive feedback and your time spent reviewing this paper.
Regarding your point "There is not enough comparison how their proposed feature gives better result compared to their previous work and other algorithms." - this paper does not give any better results compared to the previous work in terms of performance or FPGA resource costs. The benefit is in providing support of periodic tasks AND support of inter-task synchronization (via the new instructions BLOCK_TASK and UNBLOCK_TASK) AND support of 4-core CPUs. All these benefits are already described in the paper.
Regarding the spelling mistake on page 3 - (can / cane) - thank you for noticing, we corrected this mistake.
We also added extra information about the Waiting Queue and Idle Queue modules of the proposed solution + extra information about existing software-based approaches + comparison to software-based scheduling in a new section number 6 (Performance Evaluation).

Reviewer 3 Report
This paper proposes a hardware-accelerated task scheduler based on the Earliest-Deadline First (EDF) algorithm, designed as a coprocessor for real-time systems containing periodic hard real-time tasks, aiming to improve efficiency, scalability, reliability, and determinism.
Pros:
Addresses the limitations of software-based task scheduling solutions in real-time systems.
Incorporates the EDF algorithm for improved deadline-based scheduling.
Utilizes FPGA technology for flexible and efficient hardware implementation.
Introduces a set of new instructions for task management.
Cons:
Limited evaluation of the proposed solution's performance compared to existing software-based schedulers.
I recommend that the authors provide a thorough performance evaluation, comparing the proposed solution with existing software-based schedulers.
Lack of real-world use cases and application examples to demonstrate the scheduler's practical benefits.
I recommend that the authors
Include real-world use cases or application examples to demonstrate the practical benefits of the proposed scheduler.
Minor fixes:
Clarify the relationship between the different modules within the coprocessor architecture (e.g., interactions between Ready Queue, Waiting Queue, Idle Queue, etc.).
Provide more information on how the proposed scheduler handles inter-task synchronization, particularly for real-time systems with multiple tasks running simultaneously.
Elaborate on the trade-offs and limitations of using FPGA technology for implementing the scheduler, compared to other hardware acceleration techniques.
Consider discussing potential challenges and limitations of the proposed scheduler when scaling to real-time systems with a larger number of tasks or more complex task types.
Author Response
Thank you very much for your feedback and valuable comments. We did our best to address all your points to improve the paper accordingly.
"Limited evaluation of the proposed solution's performance compared to existing software-based schedulers." - we added a description of existing software-based solutions in Related Work and we added a section 6 - Performance Evaluation, where we compare the proposed solution with software approach and discuss all benefits and drawbacks.
"Lack of real-world use cases and application examples to demonstrate the scheduler's practical benefits." - this is the biggest problem to address because we do not possess any real-world use cases or application examples. But we understand that the actual application and its composition of tasks can have significant impact on the overall performance gains provided by the hardware-based scheduler. We are not denying this fact, and we did our best to describe this problem in the section 6 - Performance Evaluation (last paragraph).
Regarding Minor fixes:
"Clarify the relationship between the different modules within the coprocessor architecture (e.g., interactions between Ready Queue, Waiting Queue, Idle Queue, etc.)." - we added more information about Waiting Queue and Idle Queue in section 3.
"Provide more information on how the proposed scheduler handles inter-task synchronization, particularly for real-time systems with multiple tasks running simultaneously." - provided a paragraph in the description of Waiting Queue that explains that the actual inter-task synchronization logic still needs to be implemented in software, but the instructions BLOCK_TASK and UNBLOCK_TASK can be used by this software to block and unblock tasks. So this scheduler does not solve the synchronization problem by itself, but it is allowing to do so in software thanks to these instructions, which was not possible in our previous solutions. So this is just a support for the inter-task synchronization. Not the actual (entire) synchronization.
"Elaborate on the trade-offs and limitations of using FPGA technology for implementing the scheduler, compared to other hardware acceleration techniques." - discussed ASIC vs FPGA benefits and drawbacks in section 6 - Performance Evaluation (penultimate paragraph).
"Consider discussing potential challenges and limitations of the proposed scheduler when scaling to real-time systems with a larger number of tasks or more complex task types." - also discussed in section 6 - Performance Evaluation (third paragraph from the end).
Thank you very much your time spent reviewing this paper.
Reviewer 4 Report
The paper has been well written and provides a solution to the task scheduling problem. The article would have benefitted by having comparisons of your proposed solution with other implementations (if possible).
Author Response
Thank you very much for your positive feedback and your time.
We added a comparison with software-based scheduling in section 6 - Performance Evaluation

Round 2
Reviewer 3 Report
The Authors have addressed prior concerns.